# Dynamic Multi-Network Mining of Tensor Time Series

## ABSTRACT

The time series data pattern changes dynamically in the same way as a cluster, and so given a large collection of tensor time series consisting of multiple modes, including timestamps, how can we achieve subsequence clustering for tensor time series? In general, we do not have prior knowledge of data; hence, how can we characterize each cluster to provide interpretable insights? In this paper, we propose a new method, *Dynamic Multi-network Mining* (DMM), that converts a tensor time series into a set of segment groups of various lengths (i.e., clusters) characterized by a dependency network constrained with $\ell_1$-norm. Our method has the following properties. (a) **Interpretable**: it characterizes the cluster with multiple networks, each of which is a sparse dependency network of a corresponding non-temporal mode, and thus provides visible and interpretable insights into the key relationships. (b) **Accurate**: it discovers the clusters with distinct networks from tensor time series according to the minimum description length (MDL). (c) **Scalable**: it scales linearly in terms of the input data size when solving a non-convex problem to optimize the numbers of segments and clusters, and thus it is applicable to long-range and high-dimensional tensors. Extensive experiments with synthetic datasets confirm that our method outperforms the state-of-the-art methods in terms of clustering accuracy. We then use real datasets to demonstrate that DMM is useful for providing interpretable insights from tensor time series.

## CCS CONCEPTS

• **Information systems** → **Data mining**; *Clustering*.

## KEYWORDS

Tensor time series, Clustering, Network inference, Graphical lasso

**ACM Reference Format:**
Anonymous Author(s). 2018. Dynamic Multi-Network Mining of Tensor Time Series . In *Proceedings of Make sure to enter the correct conference title from your rights confirmation emai (Conference acronym 'XX)*. ACM, New York, NY, USA, 11 pages. https://doi.org/XXXXXXX.XXXXXXX

## 1 INTRODUCTION

The development of IoT has facilitated the collection of time series data, including data related to automobiles [27], medicine [16, 29], and finance [31, 38], from multiple modes such as sensor type, locations and users, which we call tensor time series (TTS). An instance of such data is online activity data, which records search

volumes in three modes {Query, Location, Timestamp}. These TTS can often be divided and grouped into subsequences that have similar traits (i.e., clusters). Time series subsequence clustering [1, 51] is a useful unsupervised exploratory approach for recognizing dynamic changes and uncovering interesting patterns in time series. As well as clustering data, the interpretability of the results is also important since we rarely know what each cluster refers to [33, 36]. Modeling a cluster as a dependency network [14, 40, 43], where nodes are variables and an edge expresses a relationship between variables, gives a clear explanation of what the cluster refers to. Considering that a TTS consists of multiple modes [4, 10, 23], a cluster should be modeled as multiple networks, where each is a dependency network of a corresponding non-temporal mode, to provide a good explanation. In the above example, a cluster can be modeled as query and location networks, where each explains the relationships among queries/locations. With these networks, we can understand why a particular cluster distinguishes itself from another and speculate about what happened during a period belonging to the cluster. Given such a TTS, how can we find clusters with interpretability contributing to a better understanding of the data?

Research on time series subsequence clustering has mainly focused on univariate or multivariate time series (UTS and MTS). Generally, UTS clustering methods use distance-based metrics such as dynamic time warping [5]. These methods focus on matching raw values and do not consider relationships among variables, which is essential if we are to interpret the MTS and TTS clustering. MTS clustering methods usually employ model-based clustering, which assumes, for example, a Gaussian [24] or an ARMA [47] model and attempts to find clusters that recover the data from the model. The interpretability of the clustering results depends on the model they assume. As a technique for interpretable clustering, TICC [14] models an MTS with a dependency network and discovers interpretable clusters that previously developed methods cannot find. Nevertheless, TTS clustering is a more challenging problem and cannot simply employ MTS methods due to the complexity of TTS, stemming from multiple modes, which introduces intricate dependencies and a massive data size. To employ an MTS clustering method (e.g., TICC) for TTS, the TTS must be flattened to form a higher-order MTS. As a result, the method processes the higher-order MTS and mixes up all the relationships between variables, which may capture spurious relationships and unnecessarily exacerbate the interpretability. Moreover, its computational time increases greatly as the number of variables in a mode increases.

In this paper, we propose a new method for TTS subsequence clustering, which we call *Dynamic Multi-network Mining* (DMM). [1] In our method, we define each cluster as multiple networks, each of which is a sparse dependency network of a corresponding non-temporal mode and thus can be seen as visual images that can help users quickly understand the data structure. Our algorithm scales

---

[1]Our source code and datasets are publicly available:
https://anonymous.4open.science/r/DMM-4F24.

linearly with the input data size while employing the divide-and-conquer method and is thus applicable to long-range and high-dimensional tensors. Furthermore, the clustering results and every user-defined parameter of our method can be determined by a single criterion based on the Minimum Description Length (MDL) principle [12]. DMM is a useful tool for TTS subsequence clustering that enables multifaceted analysis and understanding of TTS.

## 1.1 Preview of our results

Fig. 1 shows the DMM results for clustering over Google Trends data, which consists of 10 years of daily web search counts for six queries related to COVID-19 across 10 countries, forming a $3^{rd}$-order tensor. Fig. 1 (a) shows the cluster assignments of the TTS, where each color represents a cluster. DMM splits the tensor into four segments and groups them into four clusters, each of which can be interpreted as a distinct phase corresponding to the evolving social response to COVID-19; thus, we name these phases "*Before Covid*," "*Outbreak*," "*Vaccine*," and "*Adaptation*." It is worth noting that this result is obtained with no prior knowledge.

Fig. 1 (b) presents the networks of each cluster, i.e., a country network, which has nodes plotted on the world map, reflects dependencies between different countries, and a query network for query dependencies. These networks, also known as a Markov Random Field (MRF) [37], illustrate how the node affects the other nodes. The thickness and color of the edges in the network indicate the strength of the partial correlation between the nodes, which denotes a stronger relationship compared with a simple correlation. We learn the networks by estimating a Gaussian inverse covariance matrix. Then, by definition, if there is an edge between two nodes, the nodes are directly dependent on each other. Otherwise, they are conditionally independent, given the rest of the nodes. Moreover, we impose an $\ell_1$-norm penalty on the networks to promote sparsity, making it possible to obtain true networks and interpretability, as well as making the method noise-robust [46, 49]. These networks provide visible and interpretable insights into the key relationships that characterize clusters.

We see that each of the four clusters exhibits unique networks that evolve with the different phases. In the "*Before Covid*" phase, the country network displays edges between English-speaking countries, indicating their interconnectedness. In the query network, the query "vaccine" correlates with "influenza." However, during the "*Outbreak*" starting in 2020, many countries respond to the COVID-19 pandemic, leading to various edges in the country network. In the query network of this phase, new edges related to "coronavirus" appear, and "coronavirus" and "virus" have a particularly strong connection. In the "*Vaccine*" phase, as people become more concerned about protection from COVID-19, the query "vaccine" forms an edge with "covid." Moreover, since flu infects fewer people than in the past, "influenza" loses its edges. Lastly, during the "*Adaptation*" phase, as the world becomes accustomed to the situation, the country network reduces the number of edges, and the edges related to "influenza" reappear, reflecting a return to the networks observed in the "*Before Covid*" phase.

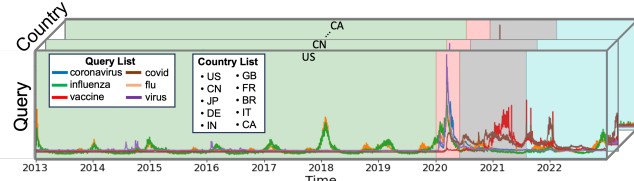

(a) Cluster assignments on the original tensor time series

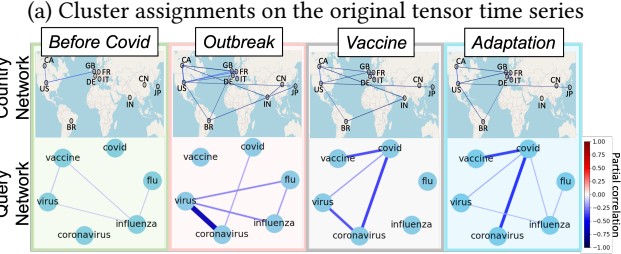

(b) Country and query networks change dynamically

**Figure 1: Effectiveness of DMM on Google Trends (#4 Covid) dataset: (a) DMM can split the tensor time series into meaningful subsequence clusters shown by colors (i.e., #green→ "*Before Covid*", #pink→ "*Outbreak*", #gray→ "*Vaccine*", #blue→ "*Adaptation*"), and (b) their important relationships between variables are summarized with country and query networks, where the nodes show individual variables, and the thickness and color of the edges are partial correlations showing the importance of its interaction.**

## 1.2 Contributions

In summary, we propose DMM as a subsequence clustering method for TTS based on the MDL principle that enables each cluster to be characterized by multiple networks. The contributions of this paper can be summarized as follows.

- **Interpretable**: DMM realizes the meaningful subsequence clustering of TTS, where each cluster is characterized by sparse dependency networks for each non-temporal mode, which facilitates the interpretation of the cluster from important relationships between variables.
- **Accurate**: We define a criterion based on MDL to discover clusters with distinct networks. Thanks to the proposed criterion, any user-defined parameters can be determined, and DMM outperforms its state-of-the-art competitors in terms of clustering accuracy on synthetic data.
- **Scalable**: The proposed clustering algorithm in DMM scales linearly as regards the input data size and is thus applicable to long-range and high-dimensional tensors.

**Outline.** The rest of the paper is organized as follows. After introducing related work in Section 2, we present our problem and basic background in Section 3. We then propose our model and algorithm in Sections 4 and 5, respectively. We report our experimental results in Sections 6 and 7.

## 2 RELATED WORK

We review previous studies that are closely related to our work.
**Time series subsequence clustering.** Subsequence clustering is an important task in time series data mining whose benefits are

the extraction of interesting patterns and the provision of valuable information, and that can also be used as a subroutine of other tasks such as forecasting [32, 39]. Time series subsequence clustering methods can be roughly separated into a distance-based method and a model-based method. The distance-based method uses metrics such as dynamic time warping [2, 5, 19] and longest common subsequence [44] and finds clusters by focusing on matching raw values rather than structure in the data. The model-based method assumes a model for each cluster, and finds the best fit of data to the model. It covers a wide variety of models such as ARMA [47], Markov chain [34], and Gaussian [24]. However, most previous work has focused on MTS and are not suitable for TTS. Few studies have focused on TTS clustering, for example, CubeScope [30] uses Dirichlet prior as a model to achieve online TTS clustering, but it only supports sparse categorical data. In summary, existing methods are not particularly well-suited to handling TTS and discovering interpretable clusters.

**Tensor time series.** TTS are ubiquitous and appear in a variety of applications, such as recommendation and demand prediction [3, 25, 45]. To model a tensor, tensor/matrix decomposition, such as Tucker/CP decomposition [21] and SVD, is a commonly used technique. Although it obtains a lower-dimensional representation that summarizes important patterns from a tensor, it struggles to capture temporal information [22]. Therefore, it is often combined with dynamical systems to handle temporal information [8, 17, 35]. For example, SSMF [18], which is an online forecasting method that uses clustering as a subroutine, combines a dynamical system with non-negative matrix factorization (NMF) to capture seasonal patterns from a TTS. Each cluster in SSMF is characterized by a lower-dimensional representation of a TTS, however, understanding the representation is demanding. Thus, tensor/matrix decomposition is not suitable for an interpretable model.

**Sparse network inference.** Inferring a sparse inverse covariance matrix (i.e., network) from data helps us to understand the dependency of variables in a statistical way. Graphical lasso [9], which maximizes the Gaussian log-likelihood imposing a $\ell_1$-norm penalty, is one of the most commonly used techniques for estimating the sparse network from static data. However, time series data are normally non-stationary, and the network varies over time; thus, to infer time-varying networks, time similarity with the neighboring network is usually considered [13]. The monitoring of such time-varying networks has been studied with the aim of analyzing economic data [31] and biological signal data [29] because of the high interpretability of the network [41]. Although the inference of time-varying networks is able to find change points by comparing the networks before and after a change, it cannot find clusters [15, 42, 48]. TICC [14] and TAGM [43] use graphical lasso and find clusters from time series based on the network of each subsequence, providing the clusters with interpretability and allowing us to discover clusters that other traditional clustering methods cannot find. However, they cannot provide an interpretable insight when dealing with TTS. Consequently, past studies have yet to find networks for TTS and to cluster TTS based on the networks. Our method uses a graphical lasso-based model modified to provide interpretable clustering results from TTS.

## 3 PROBLEM FORMULATION

**Table 1: Symbols and definitions.**

| Symbol | Definition |
|---|---|
| $D_n$ | Number of variables at mode-n |
| $N$ | Number of modes excluding temporal mode |
| $T$ | Number of timestamp |
| $\mathcal{X}$ | $(N+1)^{th}$-order TTS, i.e., $\mathcal{X} = \{\mathcal{X}_1, \mathcal{X}_2, \ldots, \mathcal{X}_T\} \in \mathbb{R}^{D_1 \times \cdots \times D_N \times T}$ |
| $\mathcal{X}_t$ | $N^{th}$-order tensor at $t^{th}$ time step, i.e., $\mathcal{X}_t \in \mathbb{R}^{D_1 \times \cdots \times D_N}$ |
| $D$ | Total product of variables excluding $T$, i.e., $D = \prod_{n=1}^{N} D_n$ |
| $D^{(\backslash n)}$ | Total product of variables excluding $D_n$ and $T$, i.e., $D^{(\backslash n)} = \prod_{m=1(m \neq n)}^{N} D_m$ |
| $K$ | Number of clusters |
| $m$ | Number of segments |
| $cp$ | Cut points, i.e., $cp = \{cp_1, cp_2, \ldots, cp_m\}$ |
| $cp_i$ | Starting point of segment $i$, i.e., $cp_1 = 1, cp_{m+1} = T+1$ |
| $\Theta$ | Model parameter set, i.e., $\Theta = \{\theta_1, \theta_2, \ldots, \theta_K\}$ |
| $\theta$ | Hierarchical Teoplitz matrix of shape $\theta \in \mathbb{R}^{D \times D}$ consists of $\{A^{(1)}, \cdots, A^{(N)}\}$ |
| $A^{(n)}$ | Precision matrix of mode-n, i.e., $A^{(n)} \in \mathbb{R}^{D_n \times D_n}$ |
| $\mathcal{F}$ | Cluster assignment set, i.e., $\mathcal{F} = \{f_1, f_2, \ldots, f_K\}$ |
| $\mathcal{M}$ | Cluster parameter set, i.e., $\mathcal{M} = \{\mathcal{F}, \Theta\}$ |
| $Cost_A(\mathcal{F})$ | Coding length cost: description complexity of $\mathcal{F}$ |
| $Cost_M(\Theta)$ | Model coding cost: description complexity of $\Theta$ |
| $Cost_C(\mathcal{X}|\mathcal{M})$ | Data coding cost: negative log-likelihood of $\mathcal{X}$ given $\mathcal{M}$ |
| $Cost_{\ell_1}(\Theta)$ | $\ell_1$-norm cost: penalty for $\Theta$ |
| $Cost_T(\mathcal{X}; \mathcal{M})$ | Total description cost: total cost of $\mathcal{X}$ given $\mathcal{M}$ |

In this section, we describe the TTS we want to analyze, introduce some necessary background material, and define the formal problem of TTS clustering.

Table 1 lists the main symbols we use throughout this paper. Consider an $(N+1)^{th}$-order TTS $\mathcal{X} \in \mathbb{R}^{D_1 \times \cdots \times D_N \times T}$, where the mode-$(N+1)$ is the time and its dimension is $T$. We can also rewrite the TTS as a sequence of $N^{th}$-order tensors $\mathcal{X} = \{\mathcal{X}_1, \mathcal{X}_2, \ldots, \mathcal{X}_T\}$, where each $\mathcal{X}_t \in \mathbf{R}^{D_1 \times \cdots \times D_N} (1 \leq t \leq T)$ denotes the observed data at the $t^{th}$ time step.

### 3.1 Tensor algebra

We briefly introduce some definitions in tensor algebra from tensor related literature [8, 21].

DEFINITION 1 (REORDER). *Let the ordered sets $P^{(1)}, \ldots, P^{(G)}$, where $P^{(g)} = \{p_1^{(g)}, \ldots, p_{n_g}^{(g)}\} \subset \{1, 2, \ldots, N\}$, be a partitioning of the modes $\{1, 2, \ldots, N\}$ s.t. $\sum_g^G n_g = N$. The reordering of an $N^{th}$-order tensor $\mathcal{X} \in \mathbf{R}^{D_1 \times \cdots \times D_N}$ into ordered sets is defined as $re(\mathcal{X})^{(P^{(1)}, \ldots, P^{(G)})} \in \mathbf{R}^{J^{(1)} \times \cdots \times J^{(G)}}$, where $J^{(g)} = \prod_{n \in P^{(g)}} D_n$.*

Given a tensor $\mathcal{X} \in \mathbf{R}^{D_1^{(1)} \times \cdots \times D_N^{(1)} \times D_1^{(2)} \times \cdots \times D_N^{(G)}}$, we partition the modes into $G$, $P^{(g)} = \{gN + 1, \cdots, g(N + 1)\}$. The element is given by $re(\mathcal{X})_{i^{(1)}, \ldots, i^{(G)}}^{(P^{(1)}, \ldots, P^{(G)})} = \mathcal{X}_{d^{(1)}, \ldots, d_N^{(1)}, d_1^{(2)}, \ldots, d_N^{(G)}}$, where $i^{(1)} = 1 + \sum_{g=1}^{N}(d_g^{(1)} - 1) \prod_{n=1}^{g-1} D_n^{(1)}$.

Special cases of reordering are vectorization and matricization. Vectorization happens when $G = 1$. $vec(\mathcal{X}) = re(\mathcal{X})^{(\{-1\})} \in \mathbf{R}^D$, where $D = \prod_{n=1}^N D_n$ and $\{-1\}$ refers to the remaining unset modes. Mode-n matricization happens when $G = 2$ and $P^{(1)}$ is a singleton. $mat(\mathcal{X})^{(n)} = re(\mathcal{X})^{(\{n\},\{-1\})} \in \mathbf{R}^{D_n \times D^{(\backslash n)}}$, where $D^{(\backslash n)} = \prod_{m=1(m \neq n)}^N D_m$.

## 3.2 Graphical lasso

We use graphical lasso as a part of our model. Given the mode-(N+1) matricization of the $(N + 1)^{th}$-order TTS, $mat(\mathcal{X})^{(N+1)} \in \mathbb{R}^{T \times D}$, the graphical lasso [9] estimates the sparse Gaussian inverse covariance matrix (i.e., network) $\theta \in \mathbb{R}^{D \times D}$, also known as the precision matrix, with which we can interpret pairwise conditional independencies among $D$ variables, e.g., if $\theta_{i,j} = 0$ then variables $i$ and $j$ are conditionally independent given the values of all the other variables. The optimization problem is given as follows:

$$\text{minimize}_{\theta \in S_{++}^p} \lambda ||\theta||_{od,1} - \sum_{t=1}^T ll(mat(\mathcal{X})_t^{(N+1)}, \theta), \quad (1)$$

$$ll(x, \theta) = -\frac{1}{2}(x - \mu)^T \theta (x - \mu)$$
$$+ \frac{1}{2} \log \det\theta - \frac{D}{2} \log(2\pi), \quad (2)$$

where $\theta$ must be a symmetric positive definite ($S_{++}^p$). $ll(x, \theta)$ is the log-likelihood and $\mu \in \mathbf{R}^D$ is the empirical mean of $mat(\mathcal{X})^{(N+1)}$. $\lambda \geq 0$ is a hyperparameter for determining the sparsity level of the network, and $|| \cdot ||_{od,1}$ indicates the off-diagonal $\ell_1$-norm. Since Eq. (1) is a convex optimization problem, its solution is guaranteed to converge to the global optimum with the alternating direction method of multipliers (ADMM) [7] and can speed up the solution time.

## 3.3 Network-based tensor time series clustering

A real-world complex $\mathcal{X}$ cannot be expressed by a single static network because it contains multiple sequence patterns, each of which has a distinct relationship/network. To address this issue, we formulate the network-based TTS clutering problem. It assumes that $T$ time steps of $\mathcal{X}$ can be divided into $m$ time segments based on $K$ networks (i.e., clusters). Let $cp$ denote a starting point set of segments, i.e., $cp = \{cp_1, cp_2, \ldots, cp_m\}$, the $i$-th segment of $\mathcal{X}$ is denoted as $\mathcal{X}_{cp_i:cp_{i+1}}$ where $cp_{m+1} = T + 1$. We group each of the $T$ points into one of the $K$ clusters denoted by a cluster assignment set $\mathcal{F} = \{f_1, f_2, \ldots, f_K\}$, where $f_k \subset \{1, 2, \ldots, T\}$, and we refer to all subsequences in the cluster $k$ as $\mathcal{X}[f_k] \subset \mathcal{X}$. Then, letting $\Theta$ be a model parameter set, i.e., $\Theta = \{\theta_1, \theta_2, \ldots, \theta_K\}$, each $\theta_k \in \mathbb{R}^{D \times D}$ is a sparse Gaussian inverse covariance matrix that summarizes the relationships of variables in $\mathcal{X}[f_k]$. Therefore, the entire cluster parameter set is given by $\mathcal{M} = \{\mathcal{M}_1, \mathcal{M}_2, \ldots, \mathcal{M}_K\}$, consisting of $\mathcal{M}_k = \{\theta_k, f_k\}$. Overall, the problem that we want to solve is written as follows.

PROBLEM 1. *Given a tensor time series $\mathcal{X}$, estimate:*
- *a cluster assignment set, $\mathcal{F} = \{f_k\}_{k=1}^K$*
- *a model parameter set, $\Theta = \{\theta_k\}_{k=1}^K$*
- *the number of clusters $K$*

## 4 PROPOSED DMM

In this section, we propose a new model with which to realize network-based TTS clustering, namely, DMM. We first describe our model $\theta$, and then we define the criterion for determining the cluster assignments and the number of clusters.

## 4.1 Multimode graphical lasso

Assume $K, \mathcal{F}$ are given, here, we address how to define and infer the model $\theta_k$. The original graphical lasso allows $\theta_k$ to connect any pairs of variables in a tensor; however, it is too high-dimensional to reveal relationships separately in terms of the non-temporal modes. To avoid the over-representation, we aim to capture the multi-aspect relationships by separating $\theta_k$ into multimodes to which we add a desired constraint for interpretability.

We assume that $\theta$ is derived from $N$ networks, $\{A^{(1)}, \ldots, A^{(N)}\}$, where $A^{(n)} \in \mathbf{R}^{D_n \times D_n}$ is the $n$-th network. For example, an element $a_{i,j}^{(n)} \in A^{(n)}$ refers to the relationship between the $i$-th and $j$-th variables of mode-n, In each network, the goal is to capture the dependencies between $D_n$ variables. We also assume that there are no relationships except among variables that differ only at mode-n. Thus, $\theta = \theta^{(N)}$ becomes an $N^{th}$ hierarchical Toeplitz matrix [11] of shape $D \times D$. $\theta^{(n)}$ can be written as follows:

$$\theta^{(n)} = \begin{pmatrix} \theta^{(n-1)} & C_{1,2}^{(n)} & \cdots & \cdots & & C_{1,D_n}^{(n)} \\ C_{2,1}^{(n)} & \theta^{(n-1)} & \cdots & & & \vdots \\ C_{3,1}^{(n)} & C_{3,2}^{(n)} & \cdots & \ddots & & \vdots \\ \vdots & \ddots & \cdots & C_{D_n-2,D_n-1}^{(n)} & C_{D_n-2,D_n}^{(n)} \\ \vdots & & \cdots & \theta^{(n-1)} & C_{D_n-1,D_n}^{(n)} \\ C_{D_n,1}^{(n)} & \ddots & \cdots & C_{D_n,D_n-1}^{(n)} & \theta^{(n-1)} \end{pmatrix},$$

where $\theta^{(1)} = A^{(1)}$ and $C_{i,j}^{(n)} \in \mathbb{R}^{D_n \times D_n}$ is a diagonal matrix whose diagonal element is $a_{i,j}^{(n)} \in A^{(n)}$, i.e., $C_{i,j}^{(n)} = a_{i,j}^{(n)} \cdot \delta_{i.j}$ allows edges that differ only at mode-n, where $\delta_{i.j}$ is the Kronecker delta.

We extend graphical lasso to obtain $\theta$ by inferring a sparse $A^{(n)}$ from a TTS. The optimization problem is written as follows:

$$\text{minimize}_{A^{(n)} \in S_{++}^p} \lambda ||A^{(n)}||_{od,1}$$
$$- \sum_t^T ll_n(re(\mathcal{X})_{t,:,:}^{(\{N+1\},\{-1\},\{n\})}, A^{(n)}), \quad (3)$$

$$ll_n(re(\mathcal{X})_{t,:,:}, A^{(n)}) = \sum_{d=1}^{D^{(\backslash n)}} \{-\frac{1}{2}(re(\mathcal{X})_{t,d,:} - \mu_d)^T A^{(n)}(re(\mathcal{X})_{t,d,:} - \mu_d)$$
$$+ \frac{1}{2} \log \det A^{(n)} - \frac{D_n}{2} \log(2\pi)\}/D^{(\backslash n)}, \quad (4)$$

where $\mu_d \in \mathbb{R}^{D_n}$ is the empirical mean of the variable $re(\mathcal{X})_{:,d,:} \in \mathbb{R}^{T \times D_n}$. Eq. (3) is a convex optimization problem solved by ADMM. We divide the log-likelihood by $D^{(\backslash n)}$ to scale the sample size.

## 4.2 Data compression

To determine the cluster assignment set $\mathcal{F}$ and the number of clusters $K$, we use the MDL principle [12], which follows the assumption

that the more we compress the data, the more we generalize its underlying structures. The goodness of the model $\mathcal{M}$ can be described with the following total description cost:

$$Cost_T(\mathcal{X}; \mathcal{M}) = Cost_A(\mathcal{F}) + Cost_M(\Theta) + \\ Cost_C(\mathcal{X}|\mathcal{M}) + Cost_{\ell_1}(\Theta). \quad (5)$$

We describe the four items that appear in Eq. (5).

**Coding length cost.** $Cost_A(\mathcal{F})$ is the description complexity of the cluster assignment set $\mathcal{F}$, which consists of the following elements: the number of clusters $K$ and segments $m$ require $\log^*(K) + \log^*(m)$. [2] The assignments of the segments to clusters require $m \times \log^*(K)$. The number of observations of each cluster requires $\sum_{k=1}^K \log^*(|f_k|)$.

$$Cost_A(\mathcal{F}) = \log^*(K) + \log^*(m) + \\ m \times \log^*(K) + \sum_{k=1}^K \log^*(|f_k|). \quad (6)$$

**Model coding cost.** $Cost_M(\Theta)$ is the description complexity of the model parameter set $\Theta$, which consists of the following elements: the diagonal values of each cluster at each hierarchy, which has sizes $D_n \times 1$, require $D_n(\log(D_n) + c_F)$, where $c_F$ is the floating point cost. [3] The positive values of $A^{(n)} \in \mathbf{R}^{D_n \times D_n}$ require $|A_k^{(n)}|_{\neq 0}(\log(D_n(D_n-1)/2) + c_F)$, where $|\cdot|_{\neq 0}$ describes the number of non-zero elements in a matrix.

$$Cost_M(\Theta) = \sum_{k=1}^K \sum_{n=1}^N \{D_n(\log(D_n) + c_F) + \log^*(|A_k^{(n)}|_{\neq 0}) + \\ |A_k^{(n)}|_{\neq 0}(\log(D_n(D_n-1)/2) + c_F)\}/(D_n^2 N). \quad (7)$$

We divide by $D_n^2 N$ to deal with the change of data scale.

**Data coding cost.** $Cost_C(\mathcal{X}|\mathcal{M})$ is the data encoding cost of $\mathcal{X}$ given the cluster parameter set $\mathcal{M}$. Huffman coding [6] uses the logarithm of the inverse of probability (i.e., the negative log-likelihood) of the values.

$$Cost_C(\mathcal{X}|\mathcal{M}) = \sum_{k=1}^K \sum_{n=1}^N \sum_{t \in f_k} ll_n(re(\mathcal{X})_{t,:,:}^{(\{N+1\},\{-1\},\{n\})}, A_k^{(n)}). \quad (8)$$

$\ell_1$**-norm cost.** $Cost_{\ell_1}(\Theta)$ is the $\ell_1$-norm cost given a model $\Theta$.

$$Cost_{\ell_1}(\Theta) = \sum_{k=1}^K \sum_{n=1}^N \lambda ||A_k^{(n)}||_{od,1}. \quad (9)$$

Discovering an optimal sparse parameter $\lambda$ capable of modeling data is a challenge, however, the parameter value can be determined by using MDL to choose the minimum total cost [26].

Our next goal is to find the best cluster parameter set $\mathcal{M}$ that minimizes the total description cost Eq. (5).

---

[2]Here, $\log^*$ is the universal code length for integers.
[3]We used $4 \times 8$ bits in our setting.

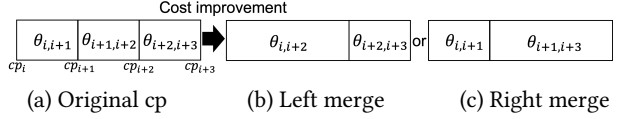

(a) Original cp  (b) Left merge  (c) Right merge

**Figure 2: Illustration of the three candidates. We compare the total description cost of each of these candidates.**

## 5 OPTIMIZATION ALGORITHMS

Thus far, we have described our model based on graphical lasso and a criterion based on MDL. The most important question is how to discover good segmentation and clustering. Here, we propose an effective and scalable algorithm. The overall procedure is summarized in Alg. 1. Given an $(N+1)^{th}$-order TTS $\mathcal{X}$, the total description cost Eq. (5) is minimized using the following two sub-algorithms.

(1) CutPointDetector: finds the number of segments $m$ and their cut points, i.e., the best cut point set $cp$ of $\mathcal{X}$.
(2) ClusterDetector: finds the number of clusters $K$ and the cluster parameter set $\mathcal{M}$.

### 5.1 CutPointDetector

The first goal is to divide a given $\mathcal{X}$ into $m$ segments (i.e., patterns), but we assume that no information is known about them in advance. Therefore, to prevent a pattern explosion when searching for their optimal cut points, we introduce CutPointDetector based on the divide-and-conquer method [20].

Specifically, it recursively merges a small segment set of $\mathcal{X}$ while reducing its total description cost, because neighboring subsequences typically exhibit the same pattern. We define $\mathbf{w}$ as a set of user-defined initial segment sizes, i.e., $\mathbf{w} = \{w_i\}_{i=1}^m$, such as the number of days in each month or any small constant. An example illustration is shown in Fig. 2. Let $\theta_{i:i+1}$ be a model of $\mathcal{X}\{cp_i : cp_{i+1}\}$ at the $i^{th}$ segment. Given the three subsequent segments illustrated in Fig. 2 (a), we evaluate whether to merge the middle segment with either of the side segments (Fig. 2 (b)(c)). The total description cost for Fig. 2 (a) is given by $Cost_T(\mathcal{X}; \{\theta_{i:i+1}, \theta_{i+1:i+2}, \theta_{i+2:i+3}\})$, where we omit the cluster assignment (e.g., $\{j\}_{j=cp_i}^{cp_{i+1}-1}$) from the cost for clarity. If the cost for the original three segments is reduced by merging, it eliminates the unnecessary cut point and employs a new model $\theta$ for the merged segment. By repeating this procedure for each segment, $m$ decreases monotonically until convergence. See Appendix A.1 for the detailed procedure.

### 5.2 ClusterDetector

Next, DMM searches for the best number of clusters by increasing $K = 1, 2, \ldots, m$, while the total description cost $Cost_T(\mathcal{X}; \mathcal{M})$ is decreasing. To compute the cost, however, we must solve two problems, namely obtain the cluster assignment set $\mathcal{F}$ and the model parameter set $\Theta$, either of which affects the optimization of the other. So, we design ClusterDetector with the expectation and maximization (EM) algorithm. In the E-step, it determines $\mathcal{F}$ to minimize the data coding cost, $Cost_C(\mathcal{X}|\mathcal{M})$, which is achieved by solving:

$$\underset{k \in \{1,\ldots,K\}}{\arg\min} \; Cost_C(\mathcal{X}|\{\theta_k, \{j\}_{j=cp_i}^{cp_{i+1}-1}\}), \quad (10)$$

---

**Algorithm 1** DMM($\mathcal{X}$, **w**)

1: **Input:** $(N+1)^{th}$-order TTS $\mathcal{X}$ and initial segment sizes set **w**
2: **Output:** Cluster parameters $\Theta$ and cluster assignments $\mathcal{F}$
3: Initialize $cp$ with **w**;
4: $cp = $ CUTPOINTDETECTOR($\mathcal{X}$, $cp$); /* Finds the best cut point set */
5: /* CLUSTERDETECTOR */
6: $K = 1$; Initialize $\Theta = \{\theta_1\}$; $\mathcal{F} = \{\{1, \ldots, T\}\}$;
7: Compute $Cost_T(\mathcal{X}; \{\Theta, \mathcal{F}\})$;
8: **repeat**
9:     $K = K + 1$; Initialize $\Theta$ for $K$ clusters;
10:     **repeat**
11:         $\mathcal{F} = $ SEGMENTASSIGNMENT($\mathcal{X}$, $\Theta$, $cp$); /* E-step */
12:         $\Theta = $ NETWORKINFERENCE($\mathcal{X}$, $\mathcal{F}$); /* M-step */
13:     **until** $\mathcal{F}$ is stable;
14:     Compute $Cost_T(\mathcal{X}; \{\Theta, \mathcal{F}\})$;
15: **until** $Cost_T(\mathcal{X}; \{\Theta, \mathcal{F}\})$ converges;
16: **return** $\mathcal{M} = \{\Theta, \mathcal{F}\}$;

---

for the $i$-th segment, and then inserts time points from $cp_i$ to $cp_{i+1}$ (i.e., $\{j\}_{j=cp_i}^{cp_{i+1}-1}$) to the best $k$-th cluster $f_k \in \mathcal{F}$. In the M-step, for $1 \leq k \leq K$ it infers $A_k^{(n)} (1 \leq n \leq N)$ according to Eq. (3) to obtain $\theta_k \in \Theta$ for a given $\mathcal{X}[f_k]$. Note that ClusterDetector starts by randomly initializing $\Theta$.

**Theoretical analysis.**

LEMMA 1. *The time complexity of DMM is $O(T \prod_{m=1}^{N} D_m)$.*

PROOF. Please see Appendix A.2. □

## 6 EXPERIMENTS

In this section, we demonstrate the effectiveness of DMM on synthetic data. We use synthetic data because there are clear ground truth networks with which to test the clustering accuracy.

### 6.1 Experimental setting

*6.1.1 Synthetic datasets.* We randomly generate synthetic $(N+1)^{th}$-order TTS, $\mathcal{X} \in \mathbb{R}^{D_1 \times \cdots \times D_N \times T}$, which follows a multivariate normal distribution $vec(\mathcal{X}_t) \sim \mathcal{N}(0, \theta^{-1})$. Each of the $K$ clusters has a mean of $\vec{0}$, so that the clustering results are based entirely on the structure of the data. For each cluster, we generate a random ground truth inverse covariance matrix $\theta$ as follows [14, 28]:

(1) For $n = 1, \ldots N$, set $A^{(n)} \in \mathbb{R}^{D_n \times D_n}$ equal to the adjacency matrix of an Erdős-Rényi directed random graph, where every edge has a 20% chance of being selected.
(2) For every selected edge in $A^{(n)}$, set $a_{i,j}^{(n)} \sim \text{Uniform}([-0.6, -0.3] \cup [0.3, 0.6])$. We enforce a symmetry constraint whereby every $a_{i,j}^{(n)} = a_{j,i}^{(n)}$.
(3) Construct a hierarchical Toeplitz matrix $\theta_{tem} \in \mathbb{R}^{D \times D}$ using $\{A^{(1)}, \cdots, A^{(N)}\}$.
(4) Let $c$ be the smallest eigenvalue of $\theta_{tem}$, and set $\theta = \theta_{tem} + (0.1 + |c|)I$, where $I$ is an identity matrix. This ensures that $\theta$ is invertible.

*6.1.2 Evaluation metrics.* We run our experiments on four different temporal sequences: A: "1,2,1", B: "1,2,3,2,1", C: "1,2,3,4,1,2,3,4",

**Table 2: Macro-F1 score of clustering accuracy for eight different temporal sequences, comparing DMM with state-of-the-art methods (higher score is better). Best results are in bold, and second best results are underlined. $^\dagger$ indicates a method where the number of clusters is set by BIC. (i): $2^{nd}$-order TTS $D_1 = 10$, (ii): $3^{rd}$-order TTS $D_1 = D_2 = 10$, A: "1,2,1", B: "1,2,3,2,1", C: "1,2,3,4,1,2,3,4", D: "1,2,2,1,3,3,3,1."**

| Data | | DMM | TAGM | TAGM $^\dagger$ | TICC | TICC $^\dagger$ |
|---|---|---|---|---|---|---|
| (i) | A | 0.955 | 0.915 | 0.915 | **0.997** | **0.997** |
| | B | **0.926** | 0.897 | 0.756 | 0.884 | 0.825 |
| | C | **0.956** | 0.770 | 0.811 | 0.725 | 0.756 |
| | D | **0.960** | 0.907 | 0.912 | 0.857 | 0.952 |
| (ii) | A | **0.961** | 0.514 | 0.514 | 0.932 | 0.923 |
| | B | **0.962** | 0.462 | 0.431 | 0.844 | 0.770 |
| | C | **0.941** | 0.359 | 0.396 | 0.704 | 0.594 |
| | D | **0.980** | 0.438 | 0.432 | 0.838 | 0.741 |

D: "1,2,2,1,3,3,3,1", (for example, A consists of three segments and two clusters $\theta_1$ and $\theta_2$.) We set each cluster in each example to have $100G$ observations, where $G$ is the number of segments in each cluster (e.g., A has $T = 300$), and cut points are set randomly. We generate each dataset ten times and report the mean of the macro-F1 score.

*6.1.3 Baselines.* We compare our method with the following two state-of-the-art methods for time series clustering using the graphical lasso as their model.

- TAGM [43]: combines HMM with a graphical lasso by modeling each cluster as a graphical lasso and assuming clusters as hidden states of HMM.
- TICC [14]: uses the Toeplitz matrix to capture lag correlations and inter-variable correlations and penalizes changing clusters to assign the neighboring segments to the same cluster.

*6.1.4 Parameter tuning.* DMM and the baselines require a sparsity parameter for $\ell_1$-norm. We varied $\lambda = \{0.5, 1, 2, 4\}$ and set $\lambda = 4$ for DMM and $\lambda = 0.5$ for the baselines, which produces the best results. A matricization of tensor $mat(\mathcal{X})^{(N+1)} \in \mathbb{R}^{T \times D}$ and the true number of clusters are given to the baselines since the number of clusters need to be set. To tune TICC, we varied the regularization parameter $\beta = \{4, 16, 64, 256\}$ and set $\beta = 16$, and set the window size $w = 1$, which is the correct assumption considering the data generation process. DMM requires us to specify **w**. We use the same $w_i$ (s.t., $i = 1, \ldots, m$) for all initial segments, and we set $w_i = 4$.

### 6.2 Results

*6.2.1 Clustering accuracy.* We take four different temporal sequences A ∼ D, and two different data sizes (i) and (ii) to observe the ability of DMM as regards clustering TTS. Table 2 shows the clustering accuracy for the macro-F1 scores for each dataset. $^\dagger$ shows TAGM and TICC set the number of clusters $K = \{2, 3, 4, 5\}$ by Bayesian information criterion (BIC). As shown, DMM outperforms the baselines in most of the datasets, even for the (i) $2^{nd}$-order TTS datasets. In particular, the difference in (ii) is even more noteworthy. This is because TAGM and TICC cannot handle $3^{rd}$-order TTS, struggling with the many variables of the matricization of the tensor.

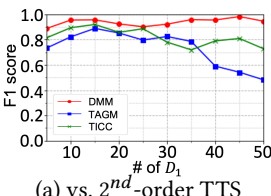

(a) vs. $2^{nd}$-order TTS

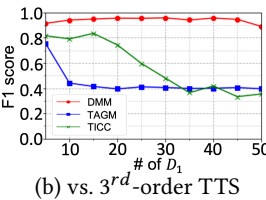

(b) vs. $3^{rd}$-order TTS

**Figure 3: DMM outperforms the state-of-the art methods: Clustering accuracy for synthetic data, macro-F1 score vs. data size, i.e., (a) $2^{nd}$-order TTS $(D_1, T) = (5 \sim 50, 800)$, (b) $3^{rd}$-order TTS $(D_1, D_2, T) = (5 \sim 50, 5, 800)$.**

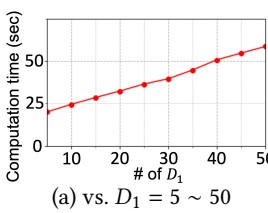

(a) vs. $D_1 = 5 \sim 50$

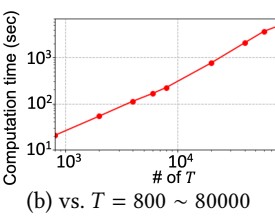

(b) vs. $T = 800 \sim 80000$

**Figure 4: DMM scales linearly: Computation time vs. data size, i.e., we vary (a) $D_1$ ($D_1 = 5 \sim 50, D_2 = 5, T = 800$) and (b) $T$ ($D_1 = 5, D_2 = 5, T = 800 \sim 80000$).**

*6.2.2 Effect of total number of variables.* We next examine how the number of variables $D_1$ affects each method as regards accurately finding clusters. We take the C example and vary $D_1 = 5 \sim 50$ for (a) $2^{nd}$-order TTS and (b) $3^{rd}$-order TTS. As shown in Fig. 3, our method outperforms the baselines for all $D_1$ in both tensors. The performance of TAGM and TICC worsens as $D_1$ increases, while DMM maintains its performance even though $D_1$ increases due to our well-defined total description cost that can handle the change in data scale. TAGM and TICC are less accurate in Fig. 3 (b) than Fig. 3 (a) since they cannot deal with $3^{rd}$-order TTS.

*6.2.3 Scalability.* We perform experiments to verify the time complexity of DMM. As described in Lemma 1, the time complexity of DMM scales linearly in terms of the data size. Fig. 4 shows the computation time of DMM when we vary $D_1$ (Fig. 4 (a)) and $T$ (Fig. 4 (b)). Thanks to our proposed optimization algorithm, the time complexity of DMM scales linearly with $D_n$ and $T$.

# 7 CASE STUDY

We perform experiments on real data to show the applicability of DMM and demonstrate how DMM can be used to obtain meaningful insights from TTS.

## 7.1 Experimental setting

*7.1.1 Datasets.* We describe our datasets in detail.
**Google Trends (#1 ∼ #5).** We use the data from Google Trends. Each tensor contains daily web-search counts. #4 Covid was collected over 10 years from Jan. 1st 2013 to Dec. 31st 2022 to include the effect of COVID-19. Other datasets are from Jan. 1st 2015 to Dec. 31st 2019 to avoid the effect of COVID-19. The datasets include five query sets (Appendix B.1). We collect the data from two target areas: three datasets from the top 10 populated US states and two from the top 10 countries ranked by GDP score. We normalize the data every month to achieve clustering that only considers the network.

**Table 3: The data size and attributes for each dataset.**

| ID | Dataset | Size | Description |
|---|---|---|---|
| #1 | E-commerce | (11, 10, 1796) | |
| #2 | VoD | (8, 10, 1796) | (query, state, day) |
| #3 | Sweets | (9, 10, 1796) | |
| #4 | Covid | (6, 10, 3652) | (query, country, day) |
| #5 | GAFAM | (5, 10, 1796) | |
| #6 | Air | (6, 12, 1461) | (pollutant, site, day) |
| #7 | Car-A | (6, 10, 4, 3241) | (sensor, lap, driver, meter) |
| #8 | Car-H | (6, 10, 4, 4000) | |

**Table 4: The number of clusters (# Cl.) and segments (# Seg.), and log-likelihood (LL) of eight real-world datasets, comparing DMM with state-of-the-art methods. The bold font and underlines show methods providing the best and second best LL, respectively (higher is better).**

| Data | # Cl. | DMM # Seg. | DMM LL | TAGM # Seg. | TAGM LL | TICC # Seg. | TICC LL |
|---|---|---|---|---|---|---|---|
| #1 | 2 | 10 | **−1.89e5** | 485 | −1.92e5 | 3 | −1.97e5 |
| #2 | 2 | 2 | −1.68e5 | 527 | **−1.65e5** | 2 | −1.68e5 |
| #3 | 2 | 7 | −1.90e5 | 502 | −1.90e5 | 17 | **−1.90e5** |
| #4 | 4 | 4 | −2.85e5 | 1778 | **−2.73e5** | 5 | −2.88e5 |
| #5 | 2 | 2 | −9.28e4 | 519 | **−9.10e4** | 3 | −9.48e4 |
| #6 | 6 | 13 | −5.19e4 | 929 | **−4.82e4** | 10 | −6.34e4 |
| #7 | 11 | 11 | **−5.89e5** | 1300 | −6.33e5 | 12 | −9.36e5 |
| #8 | 5 | 12 | **−1.06e6** | 974 | −1.02e6 | 6 | −1.16e6 |

**Air (#6).** We use Air data that collected daily concentrations of six pollutants at 12 nationally-controlled monitoring sites in Beijing, China from Mar. 1st 2013 to Feb. 29th 2016 [50]. We fill the missing values by linear interpolation and normalize the data every month.
**Automobile (#7, #8).** We use two automobile datasets with different driving courses. #7 Car-A is a city course and #8 Car-H is a highway course. We observe six sensors every meter: Brake, Speed, GX (X Accel), GY (Y Accel), Steering angle, Fuel Economy. Four drivers drive 10 laps of the same course, hence each dataset forms a $4^{th}$-order tensor. We normalize the data every 10 meters.

The size and attributes of our datasets is given in Table 3.

*7.1.2 Hyperparameter.* To tune DMM, we vary the sparsity parameter $\lambda = \{0.5, 1, 2, 4\}$ and set the value that produces the minimum total description cost. We fix the initial window size $w$ depending on the dataset, equal to the normalization period. For a fair comparison, for TAGM and TICC, we set the sparse parameter equal to DMM, and the number of clusters equal to that found by DMM. For TICC, we vary the regularization parameter $\beta = \{4, 16, 64, 256\}$ and set the parameter with BIC.

## 7.2 Results

*7.2.1 Applicability.* We show the usefulness of DMM for analyzing real-world TTS.
**Modeling acculacy.** Since there are no labels for TTS, we review the modeling accuracy of DMM by comparing the number of segments and the log-likelihood. We use cluster assignments to calculate the log-likelihood (Eq. (2)). Table 4 shows the results. DMM

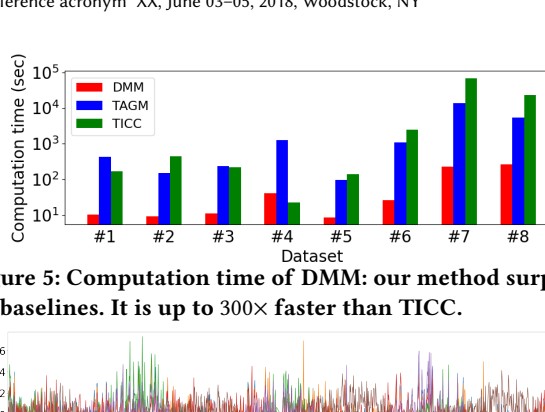

Figure 5: Computation time of DMM: our method surpasses its baselines. It is up to $300\times$ faster than TICC.

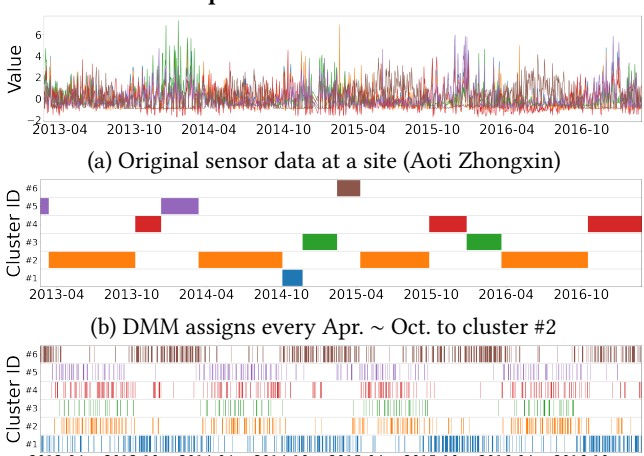

(a) Original sensor data at a site (Aoti Zhongxin)

(b) DMM assigns every Apr. ~ Oct. to cluster #2

(c) TAGM casuses frequent switching

(d) TICC assigns most periods to cluster #4

Figure 6: DMM demonstrates effective cluster assignments on the #6 Air dataset. (a) Original tensor time series data. Cluster assignments of (b) DMM, (c) TAGM and (d) TICC.

finds a reasonable number of segments and a higher log-likelihood than TICC. TAGM switches clusters with the transition matrix of HMM. This works well on synthetic datasets when there are clear transitions. However, it is not suitable for real-world datasets, which contain noises and whose network changes gradually. As a result, TAGM finds the cluster assignments that maximize the log-likelihood regardless of the number of segments. TICC assigns neighboring time steps to the same cluster using a penalty $\beta$. Thus, its number of segments is close to DMM. However, TICC is not suitable for tensors, and the log-likelihood is worse than DMM for most datasets.

**Computation time.** We compare the computation time needed for processing real data in Fig. 5. DMM is the fastest for most datasets since it infers the network for each mode. In contrast, TAGM and TICC compute the entire network at once. Therefore, they are more affected by the number of variables at each mode than DMM, resulting in a longer computation time.

*7.2.2 Interpretability.* We show how the clustering results presented by DMM make sense. We have already shown the results of DMM for clustering over #4 Covid in Section 1 (see Fig. 1). Please also see the results in #1 E-commerce in Appendix B.2.

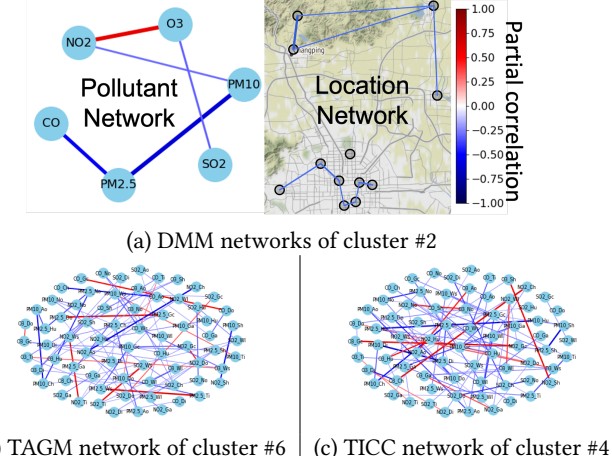

(a) DMM networks of cluster #2

(b) TAGM network of cluster #6 | (c) TICC network of cluster #4

Figure 7: Networks obtained for each methods for the #6 Air dataset: (a) DMM detects a pollutant network and a location network, where it is easy to understand the key relationships within the cluster. (b) TAGM and (c) TICC find a complex network, which is difficult to interpret.

**Air.** We compare the clustering results of DMM, TAGM and, TICC over #6 Air regarding cluster assignments (Fig. 6) and obtained networks (Fig. 7). Fig. 6 (a) shows the original sensor data at Aoti Zhongxin. Fig. 6 (b) shows that DMM assigns Apr. through Oct. of each year to cluster #2, capturing the yearly seasonality [50]. The cluster assignments of TAGM (see Fig. 6 (c)) switch frecuently, and TICC (see Fig. 6 (d)) assigns most of the period to cluster #4. Both cluster assignments are far from interpretable. Fig. 7 shows the networks obtained with each method. The cluster of DMM (see Fig. 7 (a)) includes the pollutant network and the location network. The pollutant network has a strong edge between PM2.5 and PM10, and the location network, whose nodes are plotted on the map, has edges only between closely located nodes, both of which match our expectation and accordingly indicate that DMM discovers interpretable networks. TAGM and TICC (see Fig. 7 (b)(c)) find a network for all variables. Although the networks are sparse, the large number of nodes hampers our understanding of the networks. Thus, the networks from DMM are more interpretable than with other methods. Consequently, DMM provides interpretable clustering results that can reveal underlying relationships among variables of each mode and is suitable for modeling and clustering TTS.

## 8 CONCLUSION

In this paper, we proposed an efficient tensor time series subsequence clustering method, namely DMM. Our method characterizes each cluster by multiple networks, each of which is the dependency network of a corresponding non-temporal mode. These networks make our results visible and interpretable, enabling the multifaceted analysis and understanding of tensor time series. We defined a criterion based on MDL that allows us to find clusters of data and determine all user-defined parameters. Our algorithm scales linearly with the input size and thus can apply to the massive data size of a tensor. We showed the effectiveness of DMM via extensive experiments using synthetic and real datasets.

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

**Algorithm 2** CutPointDetector$(\mathcal{X}, cp)$

1: **Input:** $(N+1)^{th}$-order TTS $\mathcal{X}$ and initial cut points set $cp$
2: **Output:** The best cut point set $cp$
3: **repeat**
4:     $id = 0, cp_{new} = \phi$;
5:     $\Theta_S = \{\theta_{cp_0:cp_1}, \theta_{cp_1:cp_2}, \ldots, \theta_{cp_m:cp_{m+1}}\}$
6:     $\Theta_E = \{\theta_{cp_0:cp_2}, \theta_{cp_2:cp_4}, \ldots\}$
7:     $\Theta_O = \{\theta_{cp_1:cp_3}, \theta_{cp_3:cp_5}, \ldots\}$
8:     **while** $id < length(\mathcal{X})$ **do**
9:       **if** $id$ is even **then**
10:        $\Theta_{Left} = \Theta_O; \Theta_{Right} = \Theta_E$;
11:        $id_{Left} = \lfloor id/2 \rfloor; id_{Right} = \lfloor id/2 \rfloor + 1$;
12:       **else if** $id$ is odd **then**
13:        $\Theta_{Left} = \Theta_E; \Theta_{Right} = \Theta_O$;
14:        $id_{Left} = \lfloor id/2 \rfloor + 1; id_{Right} = \lfloor id/2 \rfloor + 1$;
15:       **end if**
16:       $C_{solo} = Cost_T(\mathcal{X}; \{\Theta_S[id], \Theta_S[id+1], \Theta_S[id+2]\})$;
17:       $C_{left} = Cost_T(\mathcal{X}; \{\Theta_{Left}[id_{Left}], \Theta_S[id+2]\})$;
18:       $C_{Right} = Cost_T(\mathcal{X}; \{\Theta_S[id], \Theta_{Right}[id_{Right}]\})$;
19:       **if** $min(C_{solo}, C_{left}, C_{right}) = C_{solo}$ **then**
20:        $cp_{new} = cp_{new} \cup cp[id]; id += 1$;
21:       **else if** $min(C_{solo}, C_{left}, C_{right}) = C_{left}$ **then**
22:        $cp_{new} = cp_{new} \cup cp[id+1]; id += 2$;
23:       **else if** $min(C_{solo}, C_{left}, C_{right}) = C_{right}$ **then**
24:        $cp_{new} = cp_{new} \cup cp[id], cp[id+2]; id += 3$;
25:       **end if**
26:     **end while**
27:     $cp = cp_{new}$;
28: **until** $cp$ is stable;
29: **return** $cp$;

**Table 5: Google Trends query set.**

| Name | Query |
|---|---|
| #1 E-commerce | Amazon/Apple/BestBuy/Costco/Craigslist/Ebay/ Homedepot/Kohls/Macys/Target/Walmart |
| #2 VoD | AppleTV/ESPN/HBO/Hulu/Netflix/Sling/ Vudu/YouTube |
| #3 Sweets | Cake/Candy/Chocolate/Cookie/Cupcake/ Gum/Icecream/Pie/Pudding |
| #4 Covid | Covid/Corona/Flu/Influenza/Vaccine/Virus |
| #5 GAFAM | Amazon/Apple/Facebook/Google/Microsoft |

# A ALGORITHMS

## A.1 CutPointDetector

Alg. 2 shows the overall procedure for CutPointDetector, which is a subalgorithm of Alg. 1. For clarity, we describe the total description cost as $Cost_T(\mathcal{X}; \{\Theta\})$. The cluster assignment set for $\Theta[id]$ is a corresponding segment.

## A.2 Proof of Lemma 1

PROOF. The computational cost of the DMM depends largely on the number of CutPointDetector iterations and the cost of inferring $\Theta$ at each iteration. Consider that all segments are eventually merged. Since the total computational time needed to infer $\Theta$ is the

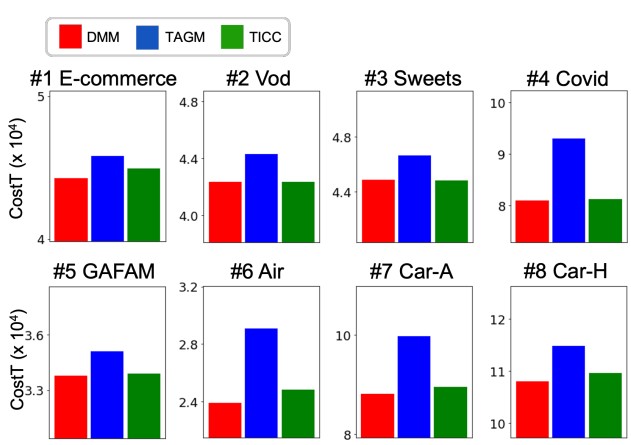

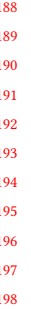

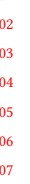

Figure 8: Total description cost of DMM: our method consistently outperforms its baselines (lower is better).

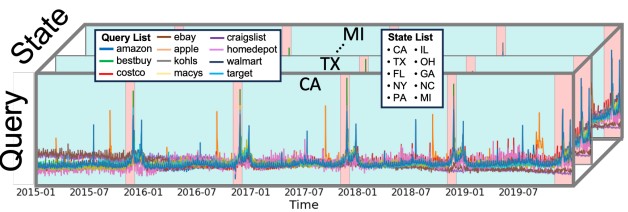

(a) Clustering results on the original tensor time series

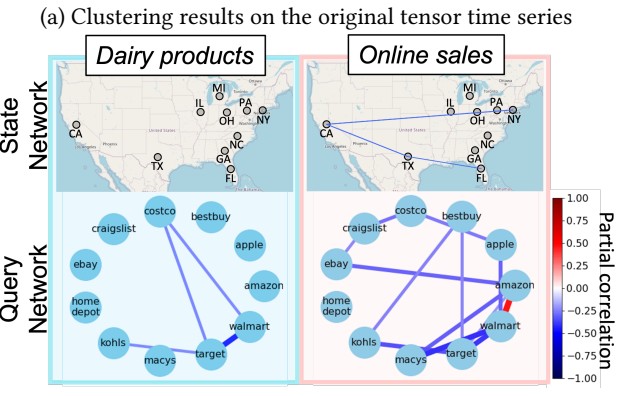

(b) State and query networks

Figure 9: Effectiveness of DMM on #1 E-commerce dataset: (a) it splits the tensor into two clusters shown by colors (i.e., #blue→ "Dairy products" and #pink→ "Online sales"). (b) each cluster has distinct state and query networks.

sum of $\{A^{(1)}, \cdots, A^{(N)}\}$ inferences, we discuss the case of $A^{(n)}$. When $T \prod_{m=1(m \neq n)}^{N} D_m \gg D_n$, at each iteration, inferring $A^{(n)}$ for all segments takes $O(D_n T \prod_{m=1(m \neq n)}^{N} D_m)$ thanks to ADMM. If the number of segments is halved at each iteration, the number of iterations is $\log_2 |\mathbf{w}|$. If the number of segments decreases by one at each iteration, the number of iterations is $|\mathbf{w}|$, but this is unlikely to happen. $T \gg \log_2 |\mathbf{w}|$, and so the computation cost related to $A^{(n)}$ is $O(T \prod_{m=1}^{N} D_m)$. Since $T, D_n \gg N$, the repetition of inference for each mode is negligible. Therefore, the time complexity of DMM is $O(T \prod_{m=1}^{N} D_m)$. □

# B CASE STUDY

## B.1 Datasets

We describe the query set we used for Google Trends in Table 5.

## B.2 Results

**Total description cost.** We compare the total description cost of DMM with TAGM and TICC on real-world datasets in Fig. 8. As shown, DMM achieves the lowest total description cost of all the datasets. TAGM has many segments, which results in the large coding length cost. TICC is not capable of handling tensor, which results in higher data coding cost compared with DMM.

**E-commerce.** We demonstrate how effectively DMM works on the #1 E-commerce dataset. Fig. 9 shows the result of DMM for clustering over #1 E-commerce. Fig. 9 (a) shows the clustering results of the original TTS, where each color represents a cluster. DMM finds 10 segments and two clusters. We name the blue cluster "Dairy products" and the pink cluster "Online sales." DMM assigns every Nov. to "Online sales", the period of Black Friday and Cyber Monday. Fig. 9 (b) shows the query and state networks for each cluster. The query network of "Daily products" shows that there are edges between the local daily products companies ("costco", "walmart", and "target"). On the other hand, with the query network of "Online sales", there are many edges, especially related to large e-commerce companies ("amazon" and "ebay"), and the state network shows that the top four populated states ("CA", "TX", "FL", and "NY") form edges, indicating the similarity of online shopping among the big states.

Received 20 February 2007; revised 12 March 2009; accepted 5 June 2009

