# OpenReview forum: "Dynamic Multi-Network Mining of Tensor Time Series"
_ACM.org/TheWebConf/2024/Conference — TheWebConf24 Oral_

### Official Review · Reviewer_gwat · 2023-11-17

**Novelty:** 5
**Technical Quality:** 5

**Review:**

**Summery:**

The paper introduces a method called Dynamic Multi-network Mining (DMM) to address the challenge of subsequence clustering for tensor time series data with multiple modes, including timestamps. The proposed method transforms tensor time series into segment groups (clusters) characterized by sparse dependency networks with l1-norm constraints. The key features of DMM include interpretability, accuracy based on minimum description length (MDL), and scalability for large and high-dimensional datasets. The method employs multiple networks for each cluster, providing visible and interpretable insights into key relationships. Experimental results on synthetic datasets demonstrate superior clustering accuracy compared to state-of-the-art methods, and real datasets illustrate the practical utility of DMM in deriving interpretable insights from tensor time series.




**Strong points:**

- Relevance to Important Topic: The paper addresses an important and timely topic, particularly given the increasing availability of data.


- Scalability with High Interpretability: The proposed approach is scalable, making it applicable to the challenges posed by growing datasets. Additionally, it offers high interpretability.


- Comprehensive Overview of Related Works: The paper offers a good overview of related works, indicating a solid understanding of the existing literature.


**Weak Points:**

- Lack of Clarity and Motivation: The paper lacks clarity and motivation. Adding more intuition and motivation could enhance the reader's understanding and engagement with the presented concepts.

- Optimality and Guarantees: Problem 1 lacks any notion of optimality and could be easily solved by using a single cluster. This limitation may weaken the practical relevance of the proposed algorithm. Additionally, the absence of guarantees for the algorithm's performance raises concerns about its robustness and reliability.

- Small Size of Used Datasets: The size of the datasets used in the experiments is considered very small. This may raise concerns about the generalizability of the proposed approach to larger, more realistic datasets.




**Minor:**

- The first question in the abstract is a bit confusing and hard to understand. Consider revising the abstract to provide a clear introduction to the paper and the addressed problem. A well-defined and easily comprehensible introduction can significantly enhance the overall clarity and accessibility of the paper.



**I read the rebuttal and it addressed my questions and concerns!**

**Questions:**

1. Could you add an optimality goal/criterion in Problem 1?

**Reviewer Confidence:**

2: The reviewer is willing to defend the evaluation, but it is likely that the reviewer did not understand parts of the paper

**Scope:**

3: The work is somewhat relevant to the Web and to the track, and is of narrow interest to a sub-community

---

### Official Review · Reviewer_zsXz · 2023-11-23

**Novelty:** 5
**Technical Quality:** 6

**Review:**

## Summary
The paper, titled "Dynamic Multi-network Mining of Tensor Time Series" (DMM), proposes a novel method for clustering tensor time series data, which consists of multiple modes including timestamps. The method is designed to convert tensor time series into segment groups (clusters) characterized by a dependency network constrained with an $\ell$1-norm. DMM aims to provide interpretability, accuracy, and scalability in analyzing large, complex tensor time series data. The method is validated through experiments on synthetic datasets and real datasets, demonstrating its effectiveness over state-of-the-art methods​​.

## Strengths
* Innovative Approach: DMM's approach to clustering tensor time series data using a multi-network model is innovative, addressing the complexity and intricacies of multiple modes within the data​​.
* Interpretability: The method focuses on interpretability, allowing for a clearer understanding of the relationships within the data, which is crucial for practical applications​​.
* Scalability: DMM is scalable and efficient, making it applicable to long-range and high-dimensional tensors, a significant advantage in handling large-scale data​​.

## Weaknesses
* Complexity of Methodology: The paper's methodology, particularly the multimode graphical lasso and the application of the MDL principle, is quite complex and may be challenging to replicate or apply in different contexts​​.
* Limited Validation on Real-World Data: While the paper includes experiments on synthetic and real datasets, more extensive validation on diverse real-world datasets is needed to establish the method's generalizability​​.
* Comparison with Existing Techniques: The paper could benefit from a more detailed comparison with existing methods, particularly in how DMM improves or differs significantly in practical applications​​.
* There are some writing errors, such as the color bar in Figure 1.

**Questions:**

## Questions
* How does DMM handle varying data distributions and potential noise in real-world datasets?
* Is there a risk of overfitting when applying DMM?
* Can the authors clarify how the MDL principle is applied in the context of DMM, particularly in terms of determining the number of clusters and segments​​?

**Reviewer Confidence:**

2: The reviewer is willing to defend the evaluation, but it is likely that the reviewer did not understand parts of the paper

**Scope:**

4: The work is relevant to the Web and to the track, and is of broad interest to the community

---

### Official Review · Reviewer_4TLC · 2023-11-29

**Novelty:** 5
**Technical Quality:** 4

**Review:**

The paper proposes a tensor time series subsequence clustering method, called DMM. DMM is able to represent each cluster with multiple networks (graphs), which are associated to the non-temporal modes of the tensor data. The graphs could be used for visualization, the interpretation of the clustering results. The proposed algorithm has linear complexity in terms of the number of timestamps, and shows fast running time compared to the baselines. Overall, the paper is well presented, with good writing, clear problem formulation, and solid theoretical analysis.

One major concern is that the experiment is only using a small selection of baselines. Furthermore, one of the only two baseline methods, TICC, as claimed by the paper in Section 7.2, is not suitable for tensors. The readers would be interested to see the comparison of the proposed method with a wider range of baselines, such as the representative but more traditional tensor decomposition method, and the more recent techniques such as the network of tensor time series, which models subsequences as networks (graphs).

One suggestion is that the paper could theoretically compare the time complexity of the proposed method with some of the representative tensor clustering algorithms, besides experimental running time comparison.

**Questions:**

The experimental results in Table 4 shows significant difference for the proposed method vs. the baselines in terms of the #Seg and LL, but the explanation is not quite clear. Please provide a more detailed analysis on the reasons that causes the difference. Also, LL seems to be not an ideal metric for this comparison. Besides LL, what might be other suitable metrics?

What are the time complexity of the baselines? From the running time comparison in Fig 5, we can not tell if the proposed method enjoys a significant time complexity advantage. Then what exactly caused the running time difference?

**Ethics Review Description:**

no ethic issues

**Reviewer Confidence:**

3: The reviewer is confident but not certain that the evaluation is correct

**Scope:**

4: The work is relevant to the Web and to the track, and is of broad interest to the community

---

### Official Review · Reviewer_Nng9 · 2023-11-29

**Novelty:** 4
**Technical Quality:** 5

**Review:**

The paper studies the problem of tensor time series (TTS) subsequence clustering. Tensor time series data referes to the time series data collected from multiple modes such as sensor types, location, and users. While time series subsequence clustering is a studied problem, TTS subsequence clustering is relatively under-explored and the topic of this paper’s investigation. The authors propose a new method for tensor time series subsequence clustering called Dynamic Multi-network Mining (DMM). DMM is a scalable, accurate and interpretable cliustering method. The clusters are characterized by multiple networks.

S1: The problem of tensor time series subsequence clustering is a relevant and important one to identify interesting trends in the data.

S2: The proposed methodology is interesting and theoretically sound.

S3: The experimental results are extensive and give a compelling argument in favor of DMM.

W1: The writing of the paper can be imporved. For example, a high level desciption of the technical methodlogy will help the reader undertsand the core technical contribution of the work better.

W2: While defining accuracy and interpretability for real-world datasets are challenging, it would have nice if the authors spent some time discussing some potential measures for capturing these metric.

**Questions:**

See W1 and W2.

**Reviewer Confidence:**

2: The reviewer is willing to defend the evaluation, but it is likely that the reviewer did not understand parts of the paper

**Scope:**

3: The work is somewhat relevant to the Web and to the track, and is of narrow interest to a sub-community

---

### Official Review · Reviewer_L8Vn · 2023-11-30

**Novelty:** 4
**Technical Quality:** 5

**Review:**

**Summary**

This paper proposes a Dynamic Multi-network Mining (DMM) model for the subsequence clustering of tensor time series. Specifically, the model uses the graph lasso to characterize the data, and uses the MDL principle for clustering. The corresponding optimization algorithm is also proposed.

The paper conducts experiments compared to methods also using the graph lasso. Results show the improved performance over baselines.

**Clarity**

Generally clear.

**Originality**

Some components of the DMM are already used in previous papers  (e.g., the use of graph lasso and MDL for clustering [14, 43, 30]), but not under the same setting as in this paper. The overall optimization algorithm is novel.

**Significance**

Pros:

1. The problem of tensor time series clustering is important and the proposed DMM model is completed.

2. The experiments validate the DMM’s accuracy, scalability and interpretability of cluster assignment.

Cons:
1. The interpretability is claimed as a main contribution of the paper. However, the criteria for interpretability is not clear to me, especially for evaluating the obtained networks. See Questions below.

2. In experiments, the baselines only include methods using the graph lasso. It is helpful to also compare with non-graph-lasso methods to see the accuracy improvement of DMM.

**Quality**

See the rating below.

**Questions:**

1. In the paper, the sparse network is preferred for interpretability. Then how to decide the sparsity for interpretability? Will the network change much under different $\lambda$? How does $\lambda$ influence the clustering accuracy?

2. In experiments, the analysis on the network  (Fig. 1 and 7) is based on visual interpretations. So maybe ‘the large number of nodes (in TICC and TAGM) hampers our understanding of the networks’. However, are there any quantitative ways to show networks from TICC and TAGM are not good for interpretation?

**Reviewer Confidence:**

2: The reviewer is willing to defend the evaluation, but it is likely that the reviewer did not understand parts of the paper

**Scope:**

4: The work is relevant to the Web and to the track, and is of broad interest to the community

---

### Official Review · Reviewer_PtBK · 2023-12-01

**Novelty:** 7
**Technical Quality:** 7

**Review:**

This paper proposes a novel approach for time subsequence clustering of tensor time series that is scalable, accurate and interpretable. Overall, the originality and technical quality of the paper is high, but the clarity could be improved.

Pros:
The paper is very technical and provides original and novel ideas.
The research is very significant since it provides a method that scales linearly and whose performance doesn't seem to degrade as the number of variables increase as compared to other state of the art methods for subsequence clustering.
The method provides interpretable and visible results that can help understand the clustering relationships between variables and be very useful for data analysis and get insights on the data.

Cons:
The paper and storyline are difficult to follow. The paper is very technical with lots of notation introduced and therefore it is understandable it is more difficult to communicate ideas but it may be good to provide more clear examples so the reader can follow along and make the storyline more easly readable. Providing examples at the introduction on how you construct the (N+1)th order TTS and the Nth order tensor for categorical examples such as #COVID dataset or for numerical/categorical examples such as #6 Air dataset would be very useful to follow your method. Clearly defining what you mean by mode and variable would also ease the reading, for instance in the case of {Query, Location, Timestamp} tensor, one of the modes is Query and its variables are the different queries in the dataset (such as covid, virus, etc). Making the definition very clear at the introduction will help the reader understand what you are trying to achieve and what you mean by dependency network. More explanations on how the covariance matrix is constructed and why is constructed this way will also help the reader, it is difficult to see which variable the columns and rows refer to in the covariance matrix.

**Questions:**

It would be useful to the reader if the authors define the difference between MTS and TTS, since MTS are mentioned many times and paper claims MTS methods cannot be used for TTS.

Can the authors clarify what they mean by hierarchical Toeplitz matrix in section 4.1 so that reader can more easily visualize it? Toeplitz matrix is supposed to have each diagonal from left to right with constant values. In the paper, each C submatrix in the covariance matrix is Toeplitz but not the whole matrix and this structure is recursively applied to diagonal submatrices.

It is not very clear or intuitive how and why the covariance matrix in section 4.1 is constructed this way. According to the definition provided, the matrix does not look DXD because every C_{i,j}^n sublock is DnXDn and there are Dn of them in a row plus the diagonal matrix which is recursive in nature. Maybe more clarification on the construction of this matrix will ease readability since this matrix is the key component on getting the interpretable networks for each mode.

The paper mentions CubeScope in the introduction for TTS clustering, but it is not used in experiments as baseline. Does the method proposed here work better for categorical data than CubeScope?

**Reviewer Confidence:**

3: The reviewer is confident but not certain that the evaluation is correct

**Scope:**

4: The work is relevant to the Web and to the track, and is of broad interest to the community

---

### Official Review · Reviewer_brwJ · 2023-12-01

**Novelty:** 4
**Technical Quality:** 4

**Review:**

This paper studies the clustering of tensor time series (TTS). The proposed method extends graphic lasso to multimode time series and adopts minimum description length as the criterion for the assignment and the number of clusters.

Pro:
This work studies an interesting topic of modeling dynamic networks in TTS and proposes a straightforward approach to model temporal dimension and non-temporal mode separately with a lower cost.

Con:
The proposed network-based TTS consists of two subproblems: temporal segmentation and clustering. Whether this two-step approach used in Alg 1 with two objectives will reach the overall optimal is questionable and less motivated except for its simplicity.

**Questions:**

- The showcase presented in Fig. 1 is very interesting. As the interpretation in temporal segmentation and clustering sometimes can be subjective, could the authors show whether DMM also has similar performance on other datasets used in Table 3?
- Could the authors provide more justification for modeling temporal and non-temporal modes separately (potentially losing their engagement) and adopting the MDL principle, and add the potential limitation of DMM?
- As Lemma 1 shows, DMM scales linearly in terms of the length of the input sequence but is exponential in the mode. Could the authors share more details on applying DMM on multimode data, especially the computation time on ID #7,8?

**Reviewer Confidence:**

3: The reviewer is confident but not certain that the evaluation is correct

**Scope:**

3: The work is somewhat relevant to the Web and to the track, and is of narrow interest to a sub-community

---

### Decision · Program_Chairs · 2024-01-22

**Decision:**

Accept (Oral)

**Comment:**

This paper introduces a novel method for subsequence clustering of tensor time series data, which is claimed to be scalable, accurate and interpretable. The method uses a sparse dependency network model with l1-norm constraints and a minimum description length criterion to cluster the data into segment groups. The paper presents experimental results on synthetic and real datasets, showing superior performance over existing methods in terms of clustering accuracy, running time and network interpretability.

 The reviews are broadly positive, and I agree with them. The paper considers a problem that is of interest to the WebConf community, and the proposed method is of sufficiently high quality. There are a few concerns raised that the authors responded to, and I recommend these are included in the final draft.